# HIV Reactivation in Latently Infected Cells with Virological Synapse-Like Cell Contact

**DOI:** 10.3390/v12040417

**Published:** 2020-04-08

**Authors:** Toshiki Okutomi, Satoko Minakawa, Riku Hirota, Koko Katagiri, Yuko Morikawa

**Affiliations:** 1Graduate School of Infection Control Sciences, Kitasato University, Shirokane 5-9-1, Minato-ku, Tokyo 108-8641, Japan; mi14003@st.kitasato-u.ac.jp (T.O.); mi17017@st.kitasato-u.ac.jp (S.M.); mi19020@st.kitasato-u.ac.jp (R.H.); 2Department of Biosciences, School of Science, Kitasato University, Kitasato 1-15-1, Minami-ku, Sagamihara, Kanagawa 252-0373, Japan; katagirk@kitasato-u.ac.jp

**Keywords:** HIV-1, latency, reactivation, cell-to-cell contact, virological synapse

## Abstract

HIV reactivation from latency is induced by cytokines but also by cell contact with other cells. To better understand this, J1.1 cells, a latent HIV-1-infected Jurkat derivative, were cocultured with its parental Jurkat. J1.1 cells became p17MA-positive and produced a high level of HIV p24CA antigen, only when they were cocultured with stimulated Jurkat with cell-to-cell contact. In contrast, very little p24CA was produced when they were cocultured without cell contact. Similar results were obtained when latent ACH-2 and its parental A3.01 cells were cocultured. Confocal microscopy revealed that not only HIV-1 p17MA and gp120Env but also LFA-1, CD81, CD59, and TCR CD3 accumulated at the cell contact site, suggesting formation of the virological synapse-like structure. LFA-1–ICAM-1 interaction was involved in the cell-to-cell contact. When J1.1 was cocultured with TCR-deficient Jurkat, the p17MA-positive rate was significantly lower, although the cell-to-cell contact was not impaired. Quantitative proteomics identified 54 membrane molecules, one of which was MHC class I, that accumulated at the cell contact site. Reactivation from latency was also influenced by the presence of stromal cells. Our study indicated that latent HIV-1 in J1.1/ACH-2 cells was efficiently reactivated by cell-to-cell contact with stimulated parental cells, accompanying the virological synapse-like structure.

## 1. Introduction

Antiretroviral therapy has slowed down the progression of HIV disease in infected individuals; however, it has revealed the presence of latent HIV-infected cells in the body [1,2,3]. Latently infected cells harbor reverse transcribed HIV cDNA but do not express HIV genes. Because of the lack of gene expression, they are resistant to anti-HIV drugs and escape the host immune systems. These latently infected cells reside in the lymphoid tissue [4,5], gut-associated lymphoid tissue [6], bone marrow [7,8], and brain [9] in HIV patients under antiretroviral therapy. They obstruct the complete cure of HIV, as the proviral HIV is reactivated when antiretroviral therapy is stopped [10,11,12]. Reactivation of latent HIV has also been observed in cases where patients have received additional IL-2 therapy [13,14].

HIV spreads via two modes of dissemination: 1) cell-free virus and 2) cell-to-cell contact [15]. The cell-to-cell transmission of HIV was observed from infected to uninfected T cells [16,17,18,19,20,21] as well as from HIV-loaded dendritic cells to T cells [22,23,24]. The cell-to-cell contact induces polarization of HIV antigens, the receptors CD4 and CXCR4/CCR5, integrins, and lipid raft markers at the cell interfaces, which form supramolecular structures resembling supramolecular activating clusters (SMAC) in immunological synapses, called virological synapses [19,23,25,26,27]. The cell-to-cell transmission of HIV is several orders of magnitude more efficient than the transmission by cell-free virus [15] and is not blocked by neutralizing antibodies or HIV fusion/entry inhibitors [17]. It is more important than previously believed, although it may be difficult to evaluate its in vivo contribution. Most of these studies have been performed using acutely infected cells, not latently infected cells. Similar studies on latently infected cells are necessary for a better understanding of cell-to-cell transmission and its contribution to HIV pathogenesis.

Numerous studies have reported that latent HIV can be reactivated by cytokines, such as TNF-α, IL-2, and IL-7 [28,29,30,31,32]. TNF-α and IL-2 are produced from mature immune cells. TNF-α is produced from macrophages and activates the NF-κB pathway, whereas IL-2 is produced from effector T cells and activates the JAK–STAT pathway. In contrast, IL-7 is produced from stromal cells present in the bone marrow and lymphoid tissue. From these facts, it can be speculated that latent HIV in peripheral organs and lymphoid tissues can be reactivated upon exposure to these cytokines. Several studies have also reported the reactivation of HIV in latently infected cells by coculture with other immune cells, such as macrophages and dendritic cells [33,34,35], and endothelial cells [36,37,38,39]. T cells, macrophages, and dendritic cells are densely populated in lymphoid tissues and transmit signals through cell-to-cell contact. For example, dendritic cells take up antigens, generate the major histocompatibility complex (MHC)-peptide complexes, and T cell receptor (TCR) recognition of peptide-MHC initiates T cell response. In such circumstances, it is plausible that latent HIV is reactivated by cell contact with surrounding lymphocytes, especially through cell contact between T cells.

Lymphocytes travel through the body from blood to lymph. They possibly encounter diverse stromal cells (e.g., fibroblasts and endothelial cells) during circulation. In lymphoid tissues, the stromal cells form a mesh network which supports hematopoiesis, and lymphocyte migration and activation [40,41]. Cell contact with and chemokines from stromal cells are potentially involved in the control and homeostasis maintenance of lymphocytes. It is possible that such stromal cells may have a direct or indirect effect on activation of provirus in latently infected T cells. A recent in vitro study has suggested that a soluble factor(s) secreted by mesenchymal stem cells reactivates latent HIV-1 [42].

In this study, we used T cell lines harboring latent HIV and cocultured them with contact or without contact to their parental T cells. Our data demonstrated that latently infected T cells efficiently converted to HIV-producing cells by direct cell-to-cell contact with stimulated T cells, leading to the formation of virological synapse-like structures. These data suggested the presence of trigger molecules on the cell surface of stimulated T cells which caused reactivation of latent provirus.

## 2. Materials and Methods

### 2.1. Cells

J1.1 (derived from Jurkat) [43] and ACH-2 (derived from A3.01) cells [44], which harbor the latent HIV-1 LAV strain and their parental Jurkat and A3.01 cells, Jβ2.7 cells (LFA-1α-deficient Jurkat cells) [45], and J.RT3-T3.5 (TCR β chain-deficient Jurkat cells) (American Type Culture Collection, Manassas, VA, USA) were grown in RPMI1640 medium supplemented with 10% fetal bovine serum (FBS). HS-5 and HS-27A cells (human stromal cell lines derived from bone marrow) (American Type Culture Collection) were grown in Dulbecco’s modified Eagle’s medium and RPMI1640 medium, supplemented with 10% FBS.

### 2.2. Cell Coculture in the Presence or Absence of Cell-to-Cell Contacts

For stimulation of uninfected cells, Jurkat and A3.01 cells were treated with 17 ng/mL phorbol 12-myristate 13-acetate (PMA) and 0.3 µM ionomycin (Sigma-Aldrich, St. Louis, MO, USA) 1 day before coculture. Stimulated cells were washed with a growth medium 3 times to remove residual PMA and ionomycin. J1.1 and ACH-2 cells were pre-labeled with 3.0 µM CellTracker Green CMFDA or Orange CMTMR (Thermo Fisher Scientific, Waltham, MA, USA) before coculture. J1.1 and ACH-2 cells were cocultured with their parental Jurkat and A3.01 cells at a 1:1 ratio (1.0 × 10^6^ cells each in 2.1 mL) in 12-well plates. For coculture without cell contact, transwell culture systems with cell culture inserts (pore size 1.0 µm, BD Falcon, Corning, NY, USA) were employed. In the transwell plates, J1.1 or ACH-2 cells were placed in the lower compartments (1.0 × 10^6^ cells in 1.5 mL) and Jurkat or A3.01 cells were placed in the upper compartments (1.0 × 10^6^ cells in 0.6 mL), respectively. In some experiments, HS-5 cells, bone marrow-derived stromal cells (1.0 × 10^5^ cells) were seeded on collagen (0.3 mg/mL, Cell Matrix, Nitta Gelatin, Osaka, Japan)-coated lower compartments of transwell plates. J1.1 cells were cocultured with Jurkat cells (2.5 × 10^5^ cells each) in the lower compartments (with contact with the HS-5 cell monolayer) or in the upper compartments (without contact with HS-5 cells). For the blocking experiments, A3.01 and ACH-2 cells were separately treated with various doses of anti-ICAM-1 mouse monoclonal antibody (mAb) (RR1/1, eBioscience, Thermo Fisher Scientific) that inhibits the ICAM-1-LFA-1 interaction [46], and were then cocultured in the presence of the mAb.

### 2.3. Monitoring of HIV-1 p24 Production and TNF-α Secretion

HIV-1 p24 production in the culture medium was measured with an HIV-1 p24CA antigen capture ELISA kit (ZeptoMetrix, Baffalo, NY, USA). In some experiments, 1.0 µM efavirenz (EFV) (Moravek Biochemicals, Brea, CA, USA) was added to the culture medium. TNF-α secretion in the culture medium was measured with a Quantikine ELISA kit (R&D Systems, Minneapolis, MN, USA).

### 2.4. Flow Cytometry (FCM)

Cells were incubated with primary mAb in phosphate-buffered saline (PBS) containing 10% FBS on ice for 30 min. The following mAbs were used in this assay: anti-CD69 mouse mAb (FN50, BD Pharmingen, Flanklin Lakes, NJ, USA), anti-LFA-1α mouse mAb (TS2/4, BioLegend, San Diego, CA, USA), anti-ICAM-1 mouse mAb (RR1/1, eBioscience), anti-CD3ε mouse mAb (HIT3a, BD Pharmingen), and anti-Flag mouse mAb (M2, Sigma-Aldrich) (as the negative control). Cells were washed with PBS containing 10% FBS and were incubated with Alexa Fluor 568-conjugated anti-mouse IgG (Invitrogen, Thermo Fisher Scientific) on ice for 30 min. After being fixed with 4% paraformaldehyde in PBS, cells were subjected to FCM with FC500 (Beckman Coulter). A total of 100,000 events were processed for each sample. Overlaid histograms were calculated by the Flowing Software developed by Perttu Terho [47].

### 2.5. Fluorescence Microscopy

Cells were placed on 0.1% (*w*/*v*) poly-L-lysine-coated coverslips and were incubated for 1 h at 37°C. Subsequently, the cells were fixed with 4% paraformaldehyde in PBS for 10 min and were permeabilized with 0.1% TritonX-100 in PBS for 10 min at room temperature. After blocking, cells were immunostained with primary Abs. The following Abs were used for immunostaining: anti-HIV-1 p17MA mouse mAb (M33-1C9, Advanced Biotechnologies, Eldersburg, MD, USA), anti-HIV-1 gp120 human polyclonal Ab (2501, Immuno Diagnostics, Woburn, MA, USA), anti-LFA-1α (CD11α) mouse mAb (TS1/22, Thermo Fisher Scientific), anti-CD59 mouse mAb (ab9183, Abcam, Cambridge, UK), anti-CD81 rabbit Ab (H-121, Santa Cruz Biotechnology, Dallas, TX, USA), anti-CD3ε mouse mAb (HIT3a, BD Pharmingen), and anti-HLA class I mouse mAb (ab23755, Abcam). Alexa Fluor 488-conjugated or Alexa Fluor 568-conjugated anti-mouse, Alexa Fluor 568-conjugated anti-human, and Alexa Fluor 647-conjugated anti-mouse IgG (Invitrogen) were used as secondary Abs. In some experiments, cells were immunostained with anti-HIV-1 p17MA mouse mAb pre-labeled with Alexa Fluor 568 by the Zenon Mouse IgG Labeling kit (Thermo Fisher Scientific). Nuclei were stained with DAPI. Cells were observed with a fluorescence microscope (BZ-8000, Keyence, Osaka, Japan) and a confocal microscope (TSC SP5, Leica, Wetzlar, Germany). Z series of optical sections performed at 0.2 to 0.5 µm intervals by confocal microscopy were subjected to reconstruction of the 3D image with image processing software Fiji in ImageJ.

### 2.6. Preparation and Analysis of Cell Surface Fractions

The isolation of cell surface proteins was performed using Pierce Cell Surface Protein Isolation Kit (Thermo Fisher Scientific). Briefly, cells were labeled with sulfo-NHS-SS-biotin, a thiol cleavable amine-reactive biotinylation reagent, and then lysed with mild detergent. Biotinylated surface proteins were isolated with avidin-agarose and were eluted using an SDS-PAGE sample buffer containing DTT [48]. The fractions (whole cell, flow through, and cell surface) were subjected to SDS-PAGE. Silver staining was performed with the Silver Stain II Kit (Wako, Osaka, Japan). Western blotting was performed with the following Abs: Anti-HSP40 rabbit Ab (ab23356, Abcam), anti-Calnexin rabbit Ab (ab22595, Abcam), anti-GAPDH mouse mAb (6C5, ambion, Thermo Fisher Scientific), anti-LFA-1α mouse mAb (27/CD11a, BD Transduction Laboratories), anti-Lck rabbit Ab (D88, Cell Signaling Technology, Danvers, MA, USA), and anti-β-actin mouse mAb (AC15, Sigma-Aldrich). The immune complexes on the membrane were visualized with the ECL prime or ECL select kit.

### 2.7. Isobaric Tag for Relative and Absolute Quantitation (iTRAQ) Analysis of Cell Surface Fractions

Proteins were isolated from the cell membrane fractions with MPEX PTS Reagents (GL Sciences). Briefly, membrane proteins (120 µg) were solubilized with MPEX reagents and digested with trypsin (SCIEX, Tokyo, Japan) at 25 °C for 20 h. After removal of MPEX reagents by solvent extraction, the resulting peptides derived from unstimulated and stimulated Jurkat cells were labeled with the iTRAQ tags 113 and 114 (iTRAQ Reagent-multiplex Assay kit, SCIEX), respectively. The labeled peptides were mixed and subjected to cation exchange chromatography. Mass spectrometry was performed using a Triple TOF 5600 system (SCIEX) and a DiNa System (KYA Technologies, Tokyo, Japan). The MS-MS spectrum data were processed using ProteinPilot Software 4.5 (SCIEX), with the Paragon algorithm in comparison with the genomic database for human (9606, UniProtKB). Protein identification was performed with a confidence threshold of a ProteinPilot Unused Score 2.0.

## 3. Results

### 3.1. Cellular Contact with Stimulated T Cells Efficiently Reactivates Latent HIV-1 Provirus

J1.1 cells, latently HIV-1-infected Jurkat cells, were cocultured with the parental Jurkat cells at a ratio of 1:1 (Figure 1). ACH-2 cells, latently HIV-1-infected A3.01 cells, were also cocultured with the parental A3.01 cells at a ratio of 1:1 (Figure 2). We temporally monitored HIV-1 p24 production from the J1.1 and ACH-2 cells by HIV-1 p24CA antigen capture ELISA (Figure 1A and Figure 2A). J1.1 cells produced low levels of p24 antigens when cultured alone, confirming previous studies [49]. When J1.1 cells were cocultured with unstimulated Jurkat cells, they showed a similar trend and produced low levels of p24. A high level of p24 production was observed when J1.1 were contact cocultured with Jurkat cells stimulated with PMA plus ionophore. However, only a low level of p24 production occurred when J1.1 and PMA/ionophore-stimulated Jurkat cells were cocultured without cell-to-cell contact in a transwell system (Figure 1A). Activation of Jurkat cells by PMA/ionophore stimulation was confirmed by upregulation of the activation marker CD69 on the Jurkat cell surface using FCM (Figure 1B). The kinetics of HIV-1 production were essentially similar when 1.0 µM EFV was added to the coculture of the J1.1 and Jurkat cells (Figure 1C). HIV-1 p24 production from acutely infected Jurkat cells was completely inhibited by the treatment with 1.0 µM EFV, while p24 was produced at 5 days post-infection in the absence of EFV (Figure 1D). These results suggested that in this cell coculture system, the HIV-1 p24 antigens observed were derived from latent J1.1 cells, but unlikely from infected Jurkat cells by transmission from J1.1 cells, at least at early time points (up to 3 days). Similar results were observed when ACH-2 cells were cocultured with either unstimulated or stimulated A3.01 cells (Figure 2A–D).

Stimulation of macrophages and T lymphocytes with PMA induces cell activation and TNF-α secretion. Several studies have shown that treatment of latently HIV-1-infected cells with TNF-α reactivates latency [33,49]. To explore this possibility, the levels of TNF-α release in these cell coculture systems were monitored by human TNF-α ELISA (Figure 1E and Figure 2E). Jurkat and A3.01 cells stimulated with PMA/ionophore produced 150–250 pg/mL of TNF-α. However, when they were washed and were cocultured with J1.1 and ACH-2 cells, little or no TNF-α was produced (Figure 1E and Figure 2E). The levels of TNF-α were insufficient for reactivation of latency in J1.1 and ACH-2 cells, which required >10 ng/mL of TNF-α (Figure 1F and Figure 2F).

To understand how latent HIV-1 in J1.1 and ACH-2 cells was reactivated by cell-to-cell contact, we performed immunostaining with anti-HIV-1 p17MA mAb that specifically detected the mature p17MA generated upon Gag processing [45]. To distinguish latently infected cells (J1.1 and ACH-2) and their counterpart parental cells (Jurkat and A3.01), J1.1 and ACH-2 cells were prelabeled with CellTracker before coculture. Fluorescence microscopy showed that p17MA antigens were visible at the plasma membranes of all coculture samples, but they were more frequently observed in coculture with stimulated Jurkat and A3.01 cells (Figure 1G and Figure 2G). For quantitation, 100–150 of CellTracker-positive J1.1 cells were observed at each time point and the ratio of p17 MA-positive J1.1 cells to the total J1.1 cells were calculated. The proportion of p17MA-positive J1.1 was high only when J1.1 were cocultured with stimulated Jurkat cells with cell-to-cell contact (Figure 1H). Careful observation, based on the identification of J1.1 cells by CellTracker prelabeling, revealed that (1) p17MA antigens were often accumulated at the contact sites of J1.1 and stimulated Jurkat cells and (2) they were rarely seen at the contact sites with unstimulated Jurkat cells (Figure 1G). The conjugation efficiencies of J1.1 with Jurkat cells were comparable and independent of the state of stimulation (Figure 1I). When quantitation was also performed for ACH-2 cells, similar findings were observed (Figure 2H), although the conjugation efficiencies of ACH-2 cells with A3.01 were reduced upon PMA/ionophore stimulation (Figure 2I). Taken together, these data suggested that although a soluble factor(s), such as cytokines, was partly responsible for the reactivation of latent HIV-1 in J1.1 and ACH-2 cells, direct cell contact with stimulated Jurkat and A3.01 cells efficiently reactivate latent HIV-1 provirus.

### 3.2. LFA-1-ICAM-1 Interaction is Involved in the Cell-to-Cell Contact, Leading to Latent HIV-1 Reactivation

The LFA-1 integrin is a well-known surface molecule which regulates cell adhesion in T cells. It binds to ICAM-1 which is expressed by antigen-presenting cells and T cells. In the immunological synapse, LFA-1 and ICAM-1 interactions are formed in the peripheral SMAC area to ensure cell conjugation. A similar organization has also been reported for virological synapse [15,19,22,23,27]. Since the J1.1-stimulated-Jurkat cell conjugation and the ACH-2-stimulated-A3.01 cell conjugation observed by us apparently resembled such synapses, we investigated whether LFA-1–ICAM-1 interactions were involved in reactivation of latent HIV-1. Jβ2.7 cells (a Jurkat cell derivative lacking LFA-1 expression) [50] were used for coculture experiments. The cell surface expression of LFA-1α and ICAM-1 was monitored by FCM (Figure 3A). LFA-1α was expressed on both unstimulated and stimulated Jurkat cells at similar levels. In contrast, LFA-1α was not expressed on Jβ2.7 cells, which was not altered upon stimulation. The LFA-1α level on J1.1 cells was considerably lower than on the parental Jurkat cells. The levels of ICAM-1 expression were similar in all the cells used in this study and did not change upon stimulation. Activation of the T cell upon PMA/ionophore stimulation was confirmed by CD69 expression on the cell surface (Figure 3A).

J1.1 cells were cocultured with stimulated Jβ2.7 cells in the presence or absence of cell-to-cell contact. Cell samples were subjected to immunostaining followed by fluorescence microscopy and the p17MA-positive rates in the J1.1 cell populations were calculated as before. The results were compared with those obtained from the coculture of the parental Jurkat and J1.1 cells (Figure 3B). The p17MA-positive J1.1 rate in the coculture with stimulated Jβ2.7 cells was slightly lower than the rate in the coculture with stimulated Jurkat cells (Figure 3B). However, the conjugation efficiency of J1.1 cells with stimulated Jβ2.7 cells did not reduce significantly and reached a level comparable to the efficiency of stimulated Jurkat cells (Figure 3C). It should be noted that in this coculture system, the LFA-1 defect was only present in Jβ2.7 cells and not in J1.1 cells, although LFA-1 expression on J1.1 cells was very low. It was possible that the LFA-1 level on J1.1 cells sufficed to slowly form conjugation with Jβ2.7 cells, allowing delayed increase of p17MA-positive rate in the J1.1–Jβ2.7 coculture system.

Next, we employed the anti-ICAM-1 mAb (RR1/1 clone) that was known to inhibit LFA-1–ICAM-1 interaction [46]. FCM analysis showed that both LFA-1 and ICAM-1 were highly expressed on unstimulated and stimulated A3.01 cells. They were expressed on ACH-2 at a lower level (Figure 3D). ACH-2 cells and simulated A3.01 cells were pretreated with either the anti-ICAM-1 mAb or control mAb (anti-FLAG mAb) and then cocultured in the presence of the mAbs. In ACH-2 cells, p17MA-positive rate decreased with anti-ICAM-1 mAb treatment in a dose-dependent fashion, however no reduction was observed in the control mAb-treated cells (Figure 3E). The conjugation efficiency of ACH-2 cells with stimulated A3.01 cells was also reduced when the cells were treated with anti-ICAM-1 mAb (Figure 3F). These results suggested that the LFA-1–ICAM-1 interaction was responsible for the cell conjugation between stimulated A3.01 and ACH-2 cells, leading to the reactivation of latent HIV-1 in ACH-2 cells. However, they are unlikely to be trigger molecules for reactivation because the expression levels were similar in unstimulated and stimulated A3.01 cells (Figure 3D), although it cannot be ruled out that HIV-1 reactivation triggered by the conformational changes of LFA-1 upon stimulation.

### 3.3. Cell-to-Cell Contact of Latently Infected T Cells and Stimulated T Cells forms Virological Synapse-Like Structure

In order to define the cell conjugates formed between the latently infected T cells and the stimulated T cells, the coculture of the stimulated Jurkat cells and CellTracker-labeled J1.1 cells was subjected to immunostaining and observed by confocal microscopy (Figure 4). HIV-1 proteins, p17MA (Figure 4A) and gp120 (Figure 4B) accumulated at the contact face of stimulated Jurkat and J1.1 cells. Similar images were observed for the conjugates between stimulated A3.01 and ACH-2 cells (Figure 4C). LFA-1 also accumulated and colocalized with the gp120 antigens at the contact site (Figure 4D). This accumulation is similar to one observed for the HIV-1 virological synapse formed by coculturing acutely infected T or dendritic cells with target T cells [23,51,52,53]. Numerous studies have shown that HIV particle assembly and budding occur at lipid raft microdomains [54,55,56] and tetraspanin-enriched microdomains (TEM) [57,58]. They were also shown to be enriched at the HIV-1 virological synapse [56,57]. Confocal analysis revealed that the lipid raft marker CD59 (Figure 4E) and the tetraspanin CD81 (Figure 4F) indeed colocalized with gp120 at the cell contact site. They often formed doughnut-like structures (Figure 4A,E). Taken together, these results suggested that the conjugates between J1.1 and stimulated Jurkat cells were formed accompanying HIV-1 virological synapse-like structures, which were composed of HIV-1 components, adhesion molecules, and lipid microdomains such as lipid rafts and TEM.

For efficiency of virological synapse formation, approximately 100 of CellTracker-positive J1.1 cells were subjected to quantitation: the conjugation efficiency of J1.1 with Jurkat cells and the gp120 accumulation at the cell-to-cell contact site (Figure 4G). The conjugation efficiencies were similar, whether or not Jurkat were stimulated. Approximately one-third of the J1.1-stimulated-Jurkat cell conjugates displayed gp120 accumulation at the cell contact site. In contrast, little or no gp120 was seen in the J1.1-unstimulated-Jurkat cell conjugation. Similar findings were observed for ACH-2 cells. Gp120 was accumulated at the contact site in one-third of the ACH-2-stimulated-A3.01 cell conjugates but was rarely seen in the cell conjugates with unstimulated A3.01 cells (Figure 4H).

### 3.4. Identification of Molecules on Stimulated T Cells which are Involved in Reactivation

TCR-CD3 is a well-known cell surface complex that transmits antigenic signals to the downstream signaling pathways for immune response. A recent study has shown that TCR in acutely infected cells is responsible for efficient HIV-1 transmission [59]. We investigated if TCR-CD3 was responsible for the reactivation of latent HIV in this study. J.RT3-T3.5 cells (TCR β chain-deficient Jurkat cells which also lack surface expression of CD3) [60,61] were utilized in coculture assays (Figure 5). FCM confirmed that CD3 was not expressed on unstimulated J.RT3-T3.5 cells, although they became slightly positive for expression after PMA/ionophore stimulation. CD69 expression was used as a marker to confirm the activation of J.RT3-T3.5 cells by PMA/ionophore stimulation. In contrast, CD3 was highly expressed on unstimulated Jurkat and sufficiently expressed, even though at slightly lower levels, on stimulated Jurkat cells (Figure 5A) [62]. J1.1 cells were cocultured with stimulated J.RT3-T3.5 cells in the presence or absence of cell-to-cell contact. Coculture samples were subjected to immunostaining and the p17MA-positive rates in J1.1 cell populations were calculated as before. The p17MA-positive J1.1 rate in coculture with stimulated J.RT3-T3.5 cells was markedly lower compared to the rate in coculture with stimulated Jurkat cells, although the rate slowly increased over time (Figure 5B). These results suggest that the TCR-CD3 complex on the signal donor cells was possibly involved in the reactivation of latent HIV-1 in target J1.1 cells. Unexpectedly, the conjugation efficiency of J1.1 cells to stimulated J.RT3-T3.5 cells was higher than that with stimulated Jurkat cells (Figure 5C). The coculture samples were immunostained for gp120 and CD3 and were subjected to confocal microscopy. CD3 as well as gp120 accumulated at the J1.1–Jurkat cell contact site. In contrast, neither CD3 nor gp120 were visible in the J1.1-J.RT3-T3.5 cell conjugation (Figure 5D). Together, these results suggest that reduction of TCR-CD3 expression on signal donor cells did not impair cell conjugation but reduced the reactivation of latent HIV-1 in J1.1 cells. More importantly, the data suggest that cell contact alone was not sufficient for the reactivation of latency in J1.1 cells and that subsequent signaling was required for the reactivation.

To identify the putative trigger molecule(s) for reactivation, cell surface proteins were purified from unstimulated and stimulated Jurkat and J1.1 cells. Western blotting with anti-HSP40 (cytosolic marker), anti-Calnexin (endoplasmic reticulum marker), and anti-GAPDH (cytosolic marker) mAbs confirmed that these cytoplasmic markers were present in the cytoplasmic fractions but almost absent in the cell surface fractions (Figure 6A). These cell fractions were subjected to silver staining. When the cell surface fractions of unstimulated and stimulated Jurkat cells were compared, several protein bands (with MW approximately 62, 59, 52, 44, 38, 32, 24, and 23 kDa) appeared with increased intensity in the stimulated cell surface fraction (shown by stars) (Figure 6B). Western blotting with anti-LFA-1 mAb revealed two glycosylated forms of LFA-1, corresponding to the mature and the immature forms, consistent with previous studies [63]. The mature form was present in the cell surface fractions but almost absent in the cytoplasmic fractions. In contrast, the immature form was enriched in the cytoplasmic fractions (Figure 6C). These results confirmed the purity of the cell surface protein fractions. Little or no mature form of LFA-1 was detected in the cell surface fraction of J1.1 cells; however, the immature form was present in the cytoplasmic fraction of J1.1 cells, suggesting that the maturation of LFA-1 was severely impaired in J1.1 cells.

We employed iTRAQ-based comparative proteomics to determine cell surface molecules upregulated due to the stimulation of Jurkat cells. The membrane proteins were recovered from unstimulated and stimulated Jurkat cells and were subjected to quantitative proteomics by tandem mass spectrometry. The iTRAQ analysis identified 1331 molecules upregulated in Jurkat cells upon stimulation (Appendix A). After exclusion of the molecules which reside in the nucleus, 54 membrane molecules, including CD69 and MHC class I antigens, were found to be significantly upregulated (Appendix A). Although this assay cannot be available for the case where the triggering of HIV reactivation occurs only by conformational changes in the molecules, this list of membrane molecules possibly includes the putative trigger molecule(s).

To explore if MHC molecules were also involved in the cell conjugates between latently infected T cells and stimulated T cells, the coculture of J1.1 cells and stimulated Jurkat cells was subjected to immunostaining and confocal microscopy (Figure 6D). HLA class I (Figure 6D) accumulated and at the contact face of stimulated Jurkat and J1.1 cells and colocalized with gp120. The HLA I accumulation was not observed in the cell conjugates between unstimulated Jurkat and J1.1 cells. Lck was not accumulated but was distributed at the cell surface and the cytoplasm.

### 3.5. Stromal Cells Contribute to Efficient Reactivation of Latent HIV-1 in T Cell Cocultures

To explore whether stromal cells were possibly involved in HIV-1 reactivation, we used HS-5 and HS-27A stromal cell lines, which are derived from bone marrow, in coculture assays. HIV-1 p24 production was monitored by HIV-1 p24CA antigen capture ELISA (Figure 7A,C,E). J1.1 cells produced a very low level of p24, when they were contact cocultured with unstimulated Jurkat cells. However, when HS-5 cells were added to the J1.1–Jurkat coculture system with cell contact, p24 production was markedly enhanced. No enhancement was seen when J1.1 cells alone were contact cocultured with HS-5 cells, suggesting that HS-5 cells did not directly reactivate latent HIV-1 in J1.1 cells. These results suggested that HS-5 cells supported the reactivation of latent HIV-1 via Jurkat cells (Figure 7A). To clarify whether this reactivation was due to a soluble factor(s) or cell-to-cell contact, HS-5 cells were separated from the J1.1-Jurkat coculture by the transwell membrane. This cell coculture system showed considerably delayed p24 production. These data were consistent with the p17MA-positive rates in J1.1 cell populations observed by fluorescence microscopy (Figure 7B). When J1.1 (prelabeled with CellTracker green) and unstimulated Jurkat cells were contact cocultured with HS-5 cells in the same compartment, the p17MA-positive rate in J1.1 cells modestly but steadily increased. However, when they were cocultured separately from HS-5 cells in a transwell system, the increase of p17MA-positive rate showed much slower kinetics. Collectively, the data suggested that both cell surface molecules and soluble factors of HS-5 cells play a potential role in the reactivation of latent HIV-1 via Jurkat cells. However, it should be noted that unlike HS-5, another stromal cell line, HS-27A cells hardly supported the reactivation of latent HIV-1 in J1.1-unstimulated Jurkat coculture system (unpublished), suggesting that this reactivation ability was stromal cell type-dependent. These observations are supported by earlier studies in which HS-27A cells secreted cytokines at a significantly lower level than HS-5 cells [64,65].

Further, we examined whether live HS-5 cells were essential for HIV reactivation in this triple coculture system. To establish this, HS-5 cells were fixed with 4% paraformaldehyde, and after extensive washings, were cocultured with J1.1 and unstimulated Jurkat cells. Neither HIV-1 particle production nor p17MA-positive cells were observed even when the cells were cocultured in the presence of cell-to-cell contact (Figure 7C,D). The results suggested that the HS-5 cell surface molecules involved in this reactivation were needed to be functional and susceptible to conformational changes.

Finally, ACH-2 cells were tested in the coculture system with HS-5 cells. A recent in vitro study has shown that the conditioned medium of mesenchymal stem cells reactivates latent HIV-1 in U1 and ACH-2 cells [43]. However, unlike the mesenchymal stem cells, little or no p24 was produced in our study when ACH-2 cells were contact cocultured with HS-5 cells. Even when ACH-2 and unstimulated A3.01 cells were cocultured and HS-5 cells were added to the ACH-2–A3.01 coculture system with cell contact, no p24 production was seen (Figure 7E). These data were consistent with the p17MA-positive rates in ACH-2 cell populations observed by fluorescence microscopy (Figure 7F). Collectively, the data suggest that although stromal cells are capable of reactivating latent HIV-1, the reactivation mechanisms also vary (by soluble factors or cell contact) and they may be altered by a combination between stromal cell types and latently infected cells.

## 4. Discussion

Previous studies showed that HIV-1 could be reactivated from latently infected T cells when they were cocultured with dendritic cells [35], macrophages [33], monocytic cells [35], or endothelial cells [36,37,38,39]. In the present study, we used T cell lines harboring latent HIV-1 provirus and their parental T cells, and showed that latent provirus was efficiently reactivated by cell-to-cell contact with the parental uninfected T cells only when they were stimulated (Figure 1 and Figure 2). Stimulation of T cells with PMA/ionomycin induces secretion of TNF-α, which is a well-known latency reversal agent. However, the levels of TNF-α produced from stimulated Jurkat/A3.01 cells (150–250 pg/mL) were approximately 50-fold lower than the dose required for reactivation in J1.1/ACH-2 cells (Figure 1E,F and Figure 2E,F). In fact, when the PMA/ionomycin-stimulated cells were washed and cocultured with latently infected cells without cell contact in a transwell system, reactivation of latent HIV-1 was not observed (Figure 1 and Figure 2).

HIV-1 reactivation from latently infected T cells was diminished when LFA-1-deficient T cells were used as signal donor cells. In addition, the inhibition of the LFA-1–ICAM-1 interaction by an antibody, which blocks cell conjugation suppressed reactivation of the latent provirus (Figure 3). These results indicated that the reactivation of latent HIV-1 was dependent on the cell conjugation mediated by LFA-1–ICAM-1 interactions.

Interestingly, confocal microscopy in this study revealed that the contact faces between latently infected cells (J1.1 and ACH-2) and stimulated T cells (Jurkat and A3.01) formed HIV-1 virological synapse-like structures (Figure 4). These observations were similar to previous reports on the coculture of acutely infected cells and target T cells [16,18,52,66]. These reports have shown that HIV-1 components, the receptors CD4 and CXCR4/CCR5, adhesion molecules such as LFA-1, and lipid raft markers and tetraspanins, are accumulated at virological synapse interfaces, and that HIV-1 particles are budded in the spatially limited area of the cell contact site [16,19,25,26,66,67]. HIV-1 virological synapse also displays distinct areas, which are somewhat similar to SMAC in immunological synapse [28]. Some studies have reported doughnut-like accumulation of HIV antigens and actin in HIV-1 virological synapses [18,19,26], suggesting that HIV components accumulate at the periphery in an early stage of virological synapse formation and converge to the center at a late stage [27,68]. Our confocal studies revealed the formation of HIV-1 virological synapse-like structures in latently infected cells. HIV-1 p17MA and gp120, LFA-1, CD59, and CD81 were accumulated at the contact site between J1.1 and uninfected Jurkat cells (Figure 4). We found that they were often observed at the periphery of the contact site, forming doughnut-like structures (Figure 4A,E). We suppose that this structure itself was not an agent for latency reversal but rather formed by the reactivation of latent HIV-1. However, it is tempting to speculate that HIV-1 reactivation through cell-to-cell contact can happen in the lymphoid tissue, where a variety of lymphocytes, such as naïve and effector cells as well as latently HIV-infected cells, are densely populated. It would be very interesting if latently infected naïve or memory T cells were contact cocultured with T cells, mimicking lymphoid microenvironments.

Our results suggest that the putative trigger molecules for HIV-1 reactivation were expressed on stimulated but not unstimulated Jurkat/A3.01 cells. Previous proteomics studies with Jurkat cells upon stimulation have reported phosphorylation in signal transduction pathways [69] and the incorporation of integral membrane and cytoskeletal proteins into lipid rafts [70]. To identify the putative trigger molecules in our cell systems, cell surface proteins were biotinylated and subsequently isolated and analyzed by SDS-PAGE. Silver staining showed that at least 8 bands (approximately 62, 59, 52, 44, 38, 32, 24, and 23 kDa) were significantly enriched in the cell surface fractions obtained from stimulated Jurkat cells (Figure 6B). Thus, our iTRAQ data will offer clues to explore putative trigger molecules for future study. A previous proteomics study has shown that cell surface expression of MHCα (approximately 44 kDa) and complement receptor-related protein (approximately 24 kDa) are upregulated in stimulated lymphocytes [71]. Our iTRAQ-based proteomics analysis also confirmed upregulation of MHC class I antigen in stimulated Jurkat cells (Appendix A). Confocal analysis revealed that HLA class I antigen accumulated at the contact site between J1.1 and stimulated Jurkat cells. This finding was not seen at the contact site with unstimulated Jurkat cells (Figure 6D).

Our confocal study also showed that the TCR CD3 accumulated at the J1.1-Jurkat contact site (Figure 5D). When we used a Jurkat derivative lacking TCR, a counterpart of MHC, in coculture experiments with J1.1 cells, HIV-1 reactivation was severely suppressed, even though the cell conjugation was not reduced (Figure 5). These results suggested that in our cell system, the TCR-CD3 complex was not required for cell conjugation but played a role in the reactivation of HIV-1. Since it has been reported that both J1.1 and ACH-2 cells severely reduce the cell surface expression of CD4 and CD3 [43,44], the reduction of TCR-CD3, a key complex of the central SMAC, not only on signal recipients but also signal donor cells, may have led to failure of TCR-MHC interactions and subsequent TCR signaling in the recipient cells. It has been debated whether the TCR–MHC interaction is involved in HIV-1 virological synapses. Early studies have suggested the absence of TCR in HIV-1 virological synapses [26,27] but a recent study has shown that TCR, although it is antigen-independent, is accumulated and required for efficient HIV-1 transmission via virological synapses [59]. From these data, the possible scenario for reactivation in our cell coculture system would be that (i) the LFA-1–ICAM-1 interaction probably functioned in forming initial cell-to-cell contact but (ii) reduced expression of CD3 in latently infected cells (signal recipient cells) did not suffice for TCR signaling unless MHC expression on signal donor cells was upregulated upon stimulation. The upregulated expression of MHC on signal donor cells may have induced enrichment of the TCR-CD3 complex at the contact site of signal recipient cells through the TCR–MHC interactions.

Lymphocytes encounter various cell types during recirculation in the body. It has been reported that HIV-1 is reactivated in latently infected cells by contact with endothelial cells [37,38,39]. Stromal cells are known to support lymphocyte migration and activation in the lymphoid tissue [40,41]. Therefore, we explored the possibility of HIV-1 reactivation from latently infected cells in the presence of stromal cells. Our results indicated that HS-5 stromal cells did not directly reactivate HIV-1 in latent J1.1 cells but supported reactivation via the Jurkat cells (Figure 7). The data also suggested that a soluble factor(s) from HS-5 cells was partly responsible for the reactivation of latent HIV-1, most likely through the stimulation of Jurkat cells. Unlike HS-5 cells, however, HS-27A cells were incapable of facilitating the reactivation (data not shown). This is consistent with the data that HS-5 cells secrete high levels of IL-1, IL-6, IL-8, IL-11, G-CSF, GM-CSF, and M-CSF, whereas HS-27A cells secrete only low levels [64,65]. In contrast, no reactivation of HIV-1 from ACH-2 cells occurred by the presence of HS-5 cells (Figure 7). However, primary mesenchymal stem cells were capable of reactivating HIV-1 in ACH-2 cells by soluble factors [42]. It is possible that primary stromal cells secrete many soluble factors, which suffice latency reactivation, as compared with stromal cell lines. Thus, various stromal cell soluble factors are needed to determine to clarify which are responsible for latency reversal.

At least 3 types of stromal cells, follicular dendritic cells (FDC), fibroblastic reticular cells (FRC), and marginal reticular cells (MRC), are localized in the lymphoid tissue [40,41]. FRC reside in the T cell zone and secrete CCL19, CCL21, CXCL16, and IL-7, whereas FDC located in the B cell follicle and MRC present in the marginal zone secrete CXCL13 [40]. It has been reported that CCL19 and CCL21 rather facilitated HIV infection and subsequent latency in resting CD4+ T cells [72,73]. The chromosomal position of the HIV genome [74,75] and its histone modification [76] are significantly associated with proviral activation. Nonetheless, the reactivation of latent provirus also depends on cues from the surrounding microenvironment of lymphoid tissues including types of T cells, stromal, and endothelial cells. Thus, trigger molecules for HIV-1 reactivation would be attractive drug targets for the prevention of HIV spreading.

## Figures and Tables

**Figure 1 viruses-12-00417-f001:**
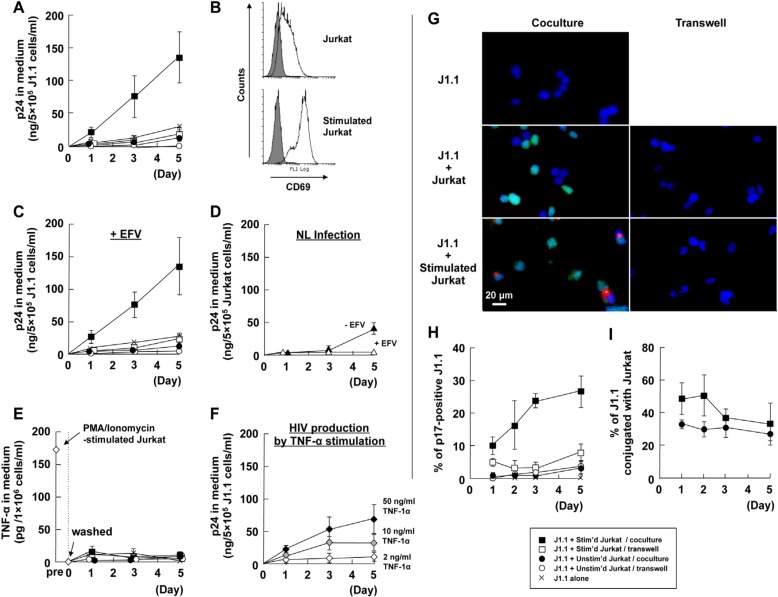
Efficient reactivation of HIV-1 in J1.1 cells by cell-to-cell contact with stimulated Jurkat cells. J1.1 cells were cocultured with unstimulated or stimulated Jurkat cells with cell-to-cell contact. J1.1 cells were also cocultured with unstimulated or stimulated Jurkat cells without cell contact in a transwell system. (**A**) HIV-1 p24 production to culture medium. The levels of HIV-1 p24 antigens were measured by HIV-1 p24CA antigen capture ELISA. Data are the mean of duplicate experiments. (**B**) CD69 expression on the cell surface. Stimulated and unstimulated Jurkat cell surfaces were immunostained with anti-CD69 mAb and were analyzed by FCM. Gray-shaded histograms show negative controls. (**C**) HIV-1 p24 production to culture medium in the presence of EFV. Cells were cocultured as described in (A) in the presence of EFV. (**D**) HIV-1 p24 production to culture medium from acutely infected Jurkat cells. Jurkat cells were infected with HIV-1 (the NL strain) at a multiplicity of infection of 2 and were cultured in the presence or absence of EFV. (**E**) TNF-α production to culture medium. The levels of TNF-α were measured by ELISA. The levels of TNF-α one day after PMA/ionomycin stimulation and subsequent cell washing were also shown. (**F**) HIV-1 p24 production by TNF-α stimulation. J1.1 cells were stimulated with TNF-α and the levels of HIV-1 p24 production were measured by HIV-1 p24CA antigen capture ELISA. (**G**) Expression of HIV-1 p17MA antigen. J1.1 cells were prelabeled with CellTracker (green) when cocultured with Jurkat cells with cell-to-cell contact. The cell mixtures were subjected to immunostaining with anti-HIV-1 p17MA mAb (red) and DAPI (blue). In the transwell cell culture, unlabeled J1.1 cells were placed in the lower compartments and Jurkat cells in the upper compartments. After coculture, the J1.1 cells in the lower compartments were immunostained. All images were at 3 days post coculture. Scale bar, 20 µm. (**H**) Reactivation efficiency of latent HIV-1 in J1.1 cells. The J1.1 cells immunostained for HIV-1 p17MA in (G) were subjected to quantification for reactivation efficiency. The rates of HIV-1 p17MA-positive J1.1 cells in 100–150 J1.1 cells were calculated. Data are the mean ± SD from three independent experiments. (**I**) Conjugation efficiency of J1.1 with Jurkat cells. The rates of J1.1 conjugated with Jurkat cells in 100–150 J1.1 cells were calculated. Data are the mean ± SD from three independent experiments.

**Figure 2 viruses-12-00417-f002:**
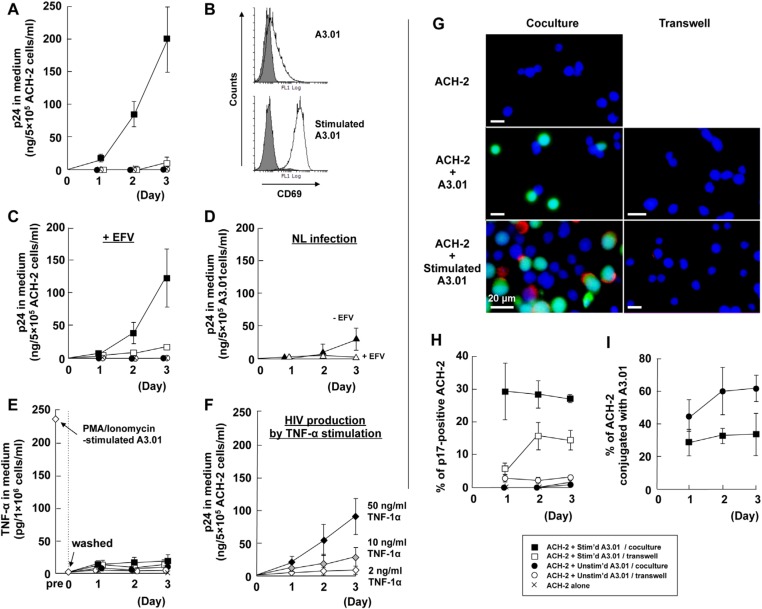
Efficient reactivation of HIV-1 in ACH2 cells by cell-to-cell contact with stimulated A3.01 cells. ACH-2 cells were cocultured with unstimulated or stimulated A3.01 cells with cell-to-cell contact. ACH-2 cells were also cocultured with unstimulated or stimulated ACH-2 cells without cell contact in a transwell system. (**A**) HIV-1 p24 production to culture medium. The levels of HIV-1 p24 antigens were measured by HIV-1 p24CA antigen capture ELISA. Data are the mean of two duplicate experiments. (**B**) CD69 expression on the cell surface. Stimulated and unstimulated A3.01 cell surfaces were immunostained with anti-CD69 mAb and analyzed by FCM. Gray-shaded histograms show negative controls. (**C**) HIV-1 p24 production to culture medium in the presence of EFV. Cells were cocultured as described in (A) in the presence of EFV. (**D**) HIV-1 p24 production to culture medium from acutely infected A3.01 cells. A3.01 cells were infected with HIV-1 (the NL strain) at a multiplicity of infection of 2 and were cultured in the presence or absence of EFV. (**E**) TNF-α production to culture medium. The levels of TNF-α were measured by ELISA. The levels of TNF-α one day after PMA/ionomycin stimulation and subsequent cell washing were also shown. (**F**) HIV-1 p24 production by TNF-α stimulation. ACH-2 cells were stimulated with TNF-α and the levels of HIV-1 p24 production were measured by HIV-1 p24CA antigen capture ELISA. (**G**) Expression of HIV-1 p17MA antigen. ACH-2 cells were prelabeled with CellTracker (green) when cocultured with A3.01 cells with cell contact. The cell mixtures were subjected to immunostaining with anti-HIV-1 p17MA mAb (red) and DAPI (blue). In the transwell cell culture, unlabeled ACH-2 cells were placed in the lower compartments and A3.01 cells in the upper compartments. The ACH-2 cells in lower compartments were immunostained. All images were at 2 days post coculture. Scale bar, 20 µm. (**H**) Reactivation efficiency of latent HIV-1 in ACH-2 cells. The ACH-2 cells immunostained for HIV-1 p17MA in (**G**) were subjected to quantification for reactivation efficiency. The rates of HIV-1 p17MA-positive ACH-2 cells in 100–150 ACH-2 cells were calculated. Data are the mean ± SD from three independent experiments. (**I**) Conjugation efficiency of ACH-2 with A3.01 cells. The rates of ACH-2 conjugated with A3.01 cells in 100–150 ACH-2 cells were calculated. Data are the mean ± SD from three independent experiments.

**Figure 3 viruses-12-00417-f003:**
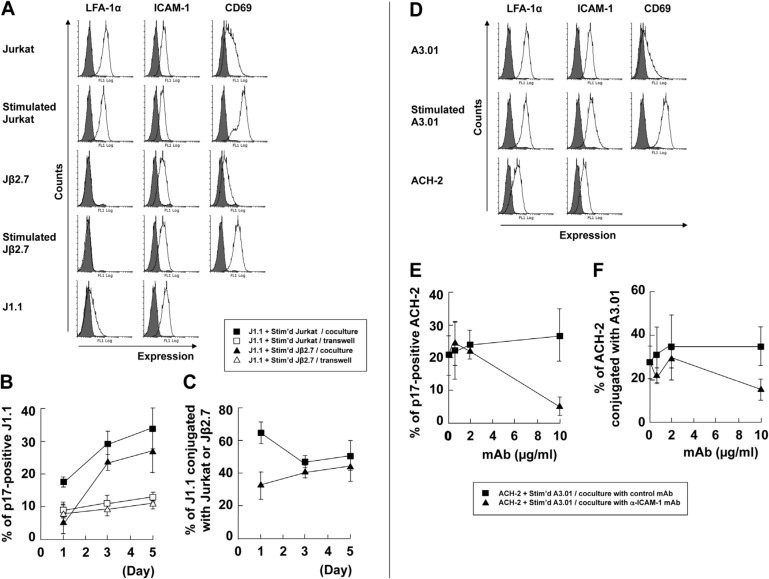
LFA-1–ICAM-1 interaction is responsible for cell conjugation with latently infected cells and reactivation of HIV-1. (**A–C**) Coculture of J1.1 cells with LFA-1α-deficient Jurkat cells. (**A**) Expression of LFA-1, ICAM-1, and CD69 on the cell surface. The cell surface of Jurkat (stimulated and unstimulated), Jβ2.7 (LFA-1α-deficient Jurkat) (stimulated and unstimulated), and J1.1 cells was immunostained with anti-LFA-1α, anti-ICAM-1, and anti-CD69 mAbs. Gray-shaded histograms show negative controls. The FCM histograms of CD69 expression in Jurkat and stimulated J1.1 cells (in Figure 1B) were shown for comparison. (**B**,**C**) J1.1 cells were cocultured with stimulated Jurkat or Jβ2.7 cells with cell-to-cell contact. J1.1 cells were similarly cocultured with stimulated Jurkat or Jβ2.7 cells without cell contact. (**B**) Reactivation efficiency of latent HIV-1 in J1.1 cells. The cells were immunostained with anti-HIV-1 p17MA mAb. The rates of p17MA-positive J1.1 cells in 100–150 J1.1 cells were calculated as described in Figure 1. (**C**) Conjugation efficiency of J1.1 with Jurkat or Jβ2.7 cells. The rates of J1.1 conjugated with Jurkat or Jβ2.7 cells in 100–150 J1.1 cells were calculated as described in Figure 1. (**D**–**F**) Coculture of ACH-2 cells in the presence of anti-ICAM-1 mAb. (**D**) LFA-1 and ICAM-1 expression on the cell surface. The cell surface of A3.01 (stimulated and unstimulated) and ACH-2 cells was stained with anti-LFA-1α and anti-ICAM-1 mAbs. Gray-shaded histograms show the negative controls. (**E**,**F**) ACH-2 and stimulated A3.01 cells were pretreated with anti-ICAM-1 mAb or control mAb and were cocultured with cell contact in the presence of the mAb for 1 day. (**E**) Reactivation efficiency of latent HIV-1 in ACH-2 cells. The rates of p17MA-positive ACH-2 cells were calculated as described in Figure 2. (**F**) Conjugation efficiency of ACH-2 cells with stimulated A3.01 cells. The rates of ACH-2 conjugated with A3.01 cells were calculated as described in Figure 2. Data are the mean ± SD from three independent experiments.

**Figure 4 viruses-12-00417-f004:**
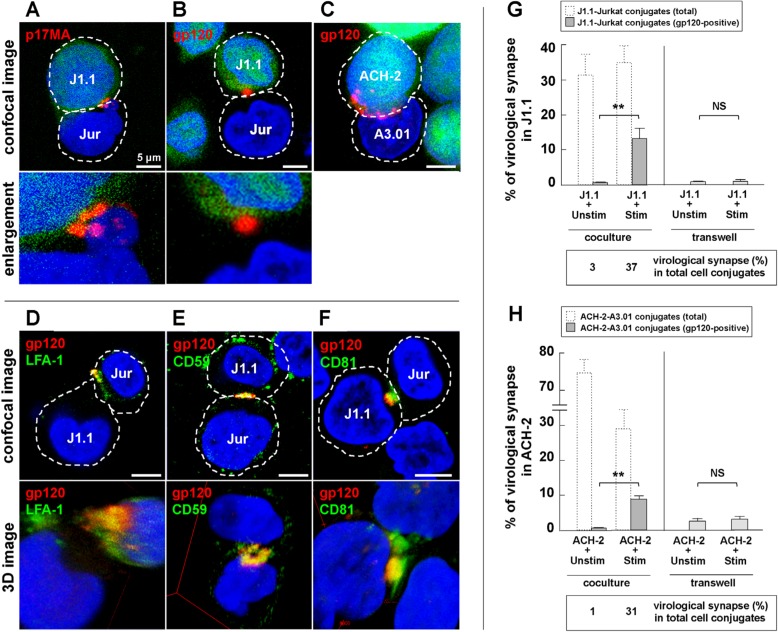
Formation of HIV-1 virological synapse-like structures at the interface of latently infected and uninfected cells. (**A,B,D–F**) Localization of HIV-1 proteins and host proteins at the J1.1–Jurkat cell interfaces. J1.1 cells were prelabeled with CellTracker and were cocultured with stimulated Jurkat cells. At 3 days post coculture, the cells were immunostained with anti-HIV-1 p17MA mAb (red), anti-HIV-1 gp120 Ab (red), anti-LFA-1α mAb (green), anti-CD59 mAb (green), and anti-CD81 mAb (green). Lower panels in (**D**–**F**) show 3D reconstruction images. (**C**) Localization of HIV-1 gp120 at ACH-2–A3.01 cell interfaces. ACH-2 cells were prelabeled with CellTracker and were cocultured with stimulated A3.01 cells. At 1 day post coculture, the cells were immunostained with anti-HIV-1 gp120 Ab (red). Nuclei were costained with DAPI (blue). Representative images were shown. Scale bar, 5 µm. (**G**) Formation of virological synapse-like structures in J1.1–Jurkat cell coculture. J1.1 cells were prelabeled with CellTracker and were cocultured with unstimulated or stimulated Jurkat cells with cell contact. At 3 days post coculture, the cells were immunostained with anti-HIV-1 gp120 Ab and were subjected to quantification for the formation of virological synapse-like structures. The rates of J1.1–Jurkat cell conjugates with HIV-1 gp120 antigens at the cell interfaces in J1.1 population were calculated (solid grey columns). The rates of J1.1–Jurkat cell conjugates in J1.1 population (100–150 J1.1 cells) were also shown (dashed columns). Efficiency of formation of virological synapse-like structures in J1.1–Jurkat cell conjugates was shown as percentage of the gp120-positive J1.1–Jurkat cell conjugates in the total J1.1–Jurkat cell conjugates (bottom panel). For comparison, J1.1 cells were cocultured with unstimulated or stimulated Jurkat cells without cell contact in a transwell system. The J1.1 cells in the lower compartments were immunostained with anti-HIV-1 gp120 Ab and the rates of HIV-1 gp120-positive J1.1 cells in 100–150 J1.1 cells were calculated. Data are the mean ± SD from three independent experiments. (**H**) Formation of virological synapse-like structures in ACH-2–A3.01 cell coculture. Cell coculture was similarly performed as described in (**G**). At 1 day post coculture, the rates of ACH-2–A3.01 conjugates in ACH-2 population (dashed columns) and the rates of ACH-2–A3.01 cell conjugates with HIV-1 gp120 antigens at the cell interface (solid grey columns) were counted. Efficiency of formation of virological synapse-like structures in ACH-2–A3.01 cell conjugates was shown as percentage of the gp120-positive ACH-2–A3.01 cell conjugates in the total J1.1–Jurkat cell conjugates (bottom panel). Data are the mean ± SD from three independent experiments. Statistical analysis was performed with a *t*-test. **, *p* < 0.01; *NS* ≥ 0.05.

**Figure 5 viruses-12-00417-f005:**
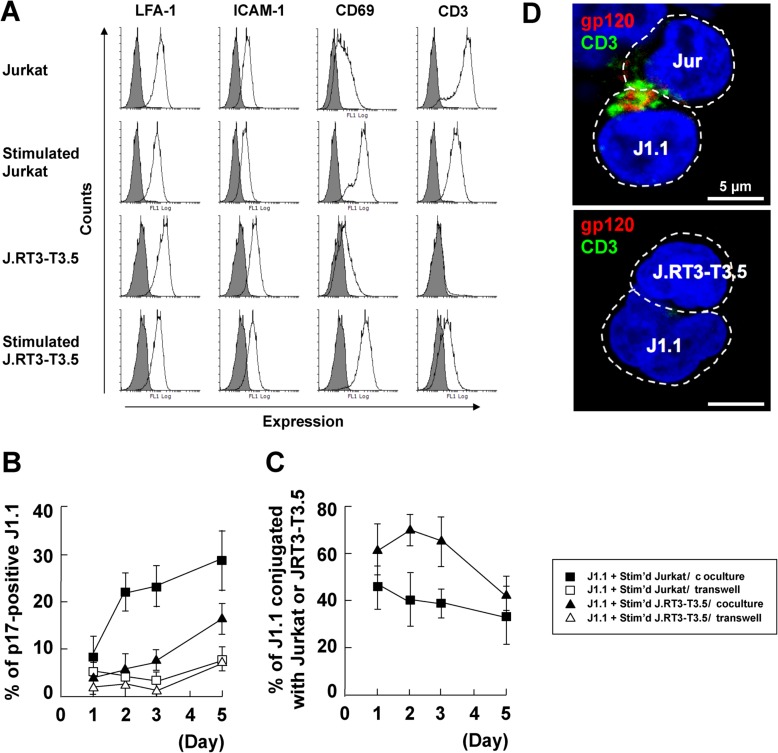
Reactivation of HIV-1 in J1.1 cells by coculture with CD3-deficient Jurkat cells. (**A**) Expression of LFA-1, ICAM-1, CD69, and CD3 on the cell surface. The cell surface of Jurkat (stimulated and unstimulated) and J.RT3-T3.5 (TCR β chain-deficient Jurkat) (stimulated and unstimulated) was immunostained with anti-LFA-1α, anti-ICAM-1, anti-CD69, and anti-CD3ε mAbs. Gray-shaded histograms show the negative controls. For comparison, the FCM data on LFA-1, ICAM-1, and CD69 expression on Jurkat cells (in Figure 3a) were shown. (**B**,**C**) J1.1 cells were cocultured with stimulated Jurkat or J.RT3-T3.5 cells with cell-to-cell contact. J1.1 cells were similarly cocultured with stimulated Jurkat or J.RT3-T3.5 cells without cell contact. (**B**) Reactivation efficiency of latent HIV-1 in the J1.1 cells. The cells were immunostained with anti-HIV-1 p17MA mAb. The rates of p17MA-positive J1.1 cells in 100–150 J1.1 cells were calculated as described in Figure 1. Data are the mean ± SD from three independent experiments. (**C**) Conjugation efficiency of J1.1 with stimulated Jurkat or J.RT3-T3.5 cells. The rates of J1.1 conjugated with Jurkat or J.RT3-T3.5 cells in 100–150 J1.1 cells were calculated as described in Figure 1. Data are the mean ± SD from three independent experiments. (**D**) J1.1 cells were prelabeled with CellTracker and were cocultured with stimulated Jurkat or J.RT3-T3.5 cells with cell contact. At 3 days post coculture, the cells were immunostained with anti-HIV-1 gp120 Ab (red) and anti-CD3ε mAb (green). Representative images were shown. Scale bar, 5 µm.

**Figure 6 viruses-12-00417-f006:**
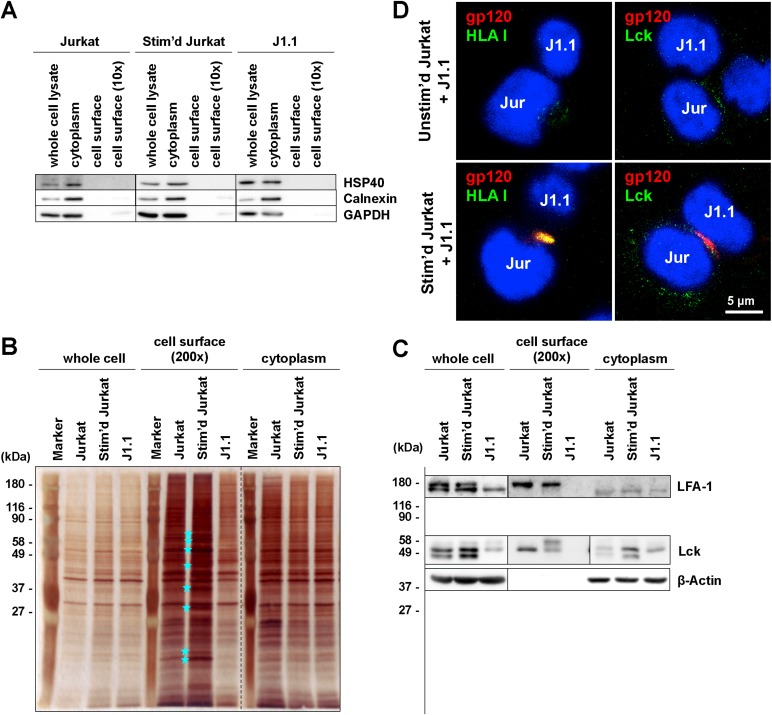
Isolation of the cell surface protein fraction and their cellular distribution. Cell surface protein fractions were isolated from the unstimulated and stimulated Jurkat cells, and J1.1 cells. The cell surface protein was biotinylated and affinity purified with avidin beads. (**A**) Marker proteins in the cell fractions isolated from the unstimulated and stimulated Jurkat cells, and the J1.1 cells. The whole cell, flow through, and affinity purified fractions were subjected to western blotting using anti-HSP40, anti-Calnexin and anti-GAPDH Abs. The 1× and 10× input were resolved on SDS-PAGE. (**B**) Silver staining of the cell fractions isolated from the unstimulated and stimulated Jurkat cells, and J1.1 cells. Stars indicate protein bands enriched in the cell surface fraction of the stimulated Jurkat cells, as compared with those of the unstimulated Jurkat cells. The 1× and 200× input were resolved on SDS-PAGE. (**C**) LFA-1 expression on the cell surface. The fractions shown in (B) were subjected to western blotting using anti-LFA-1α and anti-β-actin mAbs, and anti-Lck Ab. (**D**) J1.1 cells were prelabeled with CellTracker and were cocultured with unstimulated or stimulated Jurkat cells with cell contact. At 3 days post coculture, the cells were immunostained with anti-HIV-1 gp120 Ab (red) and anti-HLA I mAb or anti-Lck Ab (green). Representative images were shown. Scale bar, 5 µm.

**Figure 7 viruses-12-00417-f007:**
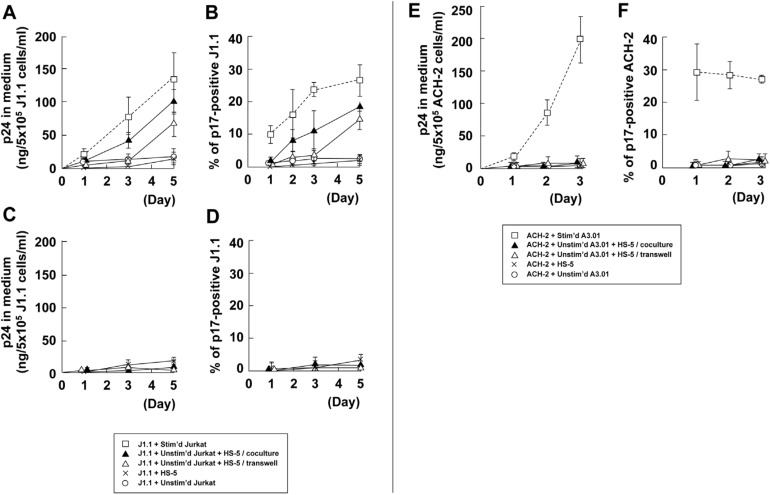
HIV-1 reactivation in the presence of stromal cells. (**A,B**) Coculture of J1.1 cells with unstimulated Jurkat in the presence of live HS-5 stromal cells. HS-5 cells were grown in the lower compartments of transwell plates. J1.1 and unstimulated Jurkat cells were added to the lower compartments (allowing cell-to-cell contact with HS-5 cells) or to the upper compartments (separately from HS-5 cells). (**C,D**) Coculture of J1.1 cells with unstimulated Jurkat in the presence of paraformaldehyde-fixed HS-5 cells. (**A,C**) HIV-1 p24 production to culture medium. The levels of HIV-1 p24 antigens were measured by HIV-1 p24CA antigen capture ELISA. (**B,D**) Reactivation efficiency of HIV-1 in J1.1 cells. J1.1 cells (prelabeled) and unstimulated Jurkat cells were cocultured in the presence of HS-5 cells, and the J1.1–Jurkat cell mixtures were subjected to immunostaining for HIV-1 p17MA. The p17MA-positive rates were calculated as described in Figure 1. Data are the mean ± SD from three independent experiments. For comparison, the data on the J1.1-stimulated Jurkat coculture with cell-to-cell contact (in Figure 1A,H) were shown (dashed lines). (**E,F**) Coculture of ACH-2 cells with unstimulated A3.01 in the presence of live HS-5 stromal cells. HS-5 cells were grown in lower compartments of transwell plates, and ACH-2 and unstimulated A3.01 cells were cocultured with cell contact in the lower compartments or separately in the upper compartments. (**E**) HIV-1 p24 production to culture medium. (**F**) Reactivation efficiency of HIV-1 in ACH-2 cells. The p17MA-positive rates were calculated as described in Figure 2. Data are the mean ± SD from three independent experiments. For comparison, the data on the ACH-2-stimulated A3.01 coculture with cell-to-cell contact (in Figure 2A,H) were shown (dashed lines).

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
