# Peer review of "HIV Reactivation in Latently Infected Cells with Virological Synapse-Like Cell Contact"

_viruses, 2020, doi:10.3390/v12040417_

Round 1

Reviewer 1 Report

The article, “HIV reactivation in latently infected cells with virological synapse-like cell contact” is a solid piece of science in an important area of HIV Cure research.  In this paper, the authors present data showing that although a soluble factor(s) such as cytokines were partly responsible for reactivation of latent HIV-1 in J1.1 and ACH-2 cells, direct cell contact with stimulated Jurkat and A3.01 cells efficiently reactivated latent HIV-1 provirus. LFA-1-ICAM-1 interaction was responsible for the cell conjugation between stimulated A3.01 and ACH-2 cells, leading to reactivation of latent HIV-1 in ACH-2 cells. The conjugates between J1.1 and stimulated Jurkat cells were formed accompanying HIV-1 virological synapse-like structures, which were composed of HIV-1 components, adhesion molecules, and lipid microdomains such as lipid rafts and TEM. J.RT3-T3.5 cells. TCR-CD3 complex on signal donor cells was possibly involved in reactivation of latent HIV-1 in target J1.1 cells. A reduction of TCR-CD3 expression on signal donor cells did not impair cell conjugation but reduced the reactivation of latent HIV-1 in J1.1 cells. To identify the putative trigger molecule(s) for reactivation iTRAQ-based comparative proteomics were used to determine cell surface molecules upregulated due to stimulation of Jurkat cells. 54 membrane molecules, including CD69 and MHC class I antigens, were found to be significantly upregulated. HLA class I accumulated and at the contact face of stimulated Jurkat and J1.1 cells and colocalized with gp120. The HS-5 cells supported the reactivation of latent HIV-1 via Jurkat cells (that both cell surface molecules and soluble factors of HS-5 cells played a potential role in the reactivation of latent HIV-1 via Jurkat cells. However, not all stromal cells behaved the same: unlike HS-5, another stromal cell line, HS-27A cells, hardly supported the reactivation of  that although stromal cells are capable of reactivating latent HIV-1.  The reactivation mechanisms also vary (by soluble factors or cell contact) and they may be altered by a combination between stromal cell types and latently infected cells.

The following are some points to consider to make the paper more clear and impactful.

Line 38: it is cure of HIV rather than the disease AIDS. replace AIDS with HIV

Line 48: with respect to the use of “anti-HIV drugs”: be more specific here. Fusion and entry inhibitors do not affect cell cell contact but RT and integrase inhibitors should

Line 337: Please explain why bone marrow derived stromal cells are relevant to HIV latency. Shouldn’t these be derived from lymphatic tissue?

Figure 7: it might be useful to a)  look at the effect of HS-5 on Jurkat stimulation and b) determine some of the soluble factors using depletion of HS-5 conditioned media using monoclonal antibodies to various factors.

Line 616: The further determination of the various stromal cell soluble/secreted factors should be followed-up.

Line 632: Please explain the last sentence:  “target molecules to inhibit?” I don’t understand this point. Perhaps explain more and expand.

Author Response

We appreciate the reviewer to consider our study solid. The reviewer kindly suggested the following points:

1) Line 38: it is cure of HIV rather than the disease AIDS. replace AIDS with HIV.

R1) We replaced “AIDS” with “HIV”.

2) Line 48: with respect to the use of “anti-HIV drugs”: be more specific here. Fusion and entry inhibitors do not affect cell cell contact but RT and integrase inhibitors should.

R2) “anti-HIV drugs” was changed to “HIV fusion/entry inhibitors”.

3) Line 337: Please explain why bone marrow derived stromal cells are relevant to HIV latency. Shouldn’t these be derived from lymphatic tissue?

R3) As stromal cell lines derived from human lymphatic tissues were not commercially available, we instead used bone marrow-derived stromal cell lines.

4) Figure 7: it might be useful to a) look at the effect of HS-5 on Jurkat stimulation and b) determine some of the soluble factors using depletion of HS-5 conditioned media using monoclonal antibodies to various factors.

R4) (a) We are so glad to know that the reviewer supports the possibility of HS-5 effect on Jurkat stimulation. We agree the possibility and have suggested it in the text (line 600-601). (b) We think that HS-5 stimulate Jurkat through soluble factors as well as cell contact. We are working on the soluble factors responsible for Jurkat stimulation, but depletion of the factors by adding their specific antibodies to the cell coculture failed so far, because the antibodies were degraded or endocytosed in the coculture period (~5 days). We may need to establish an experimental system in which Jurkat is stimulated by HS-5 conditioned medium alone.

5) Line 616: The further determination of the various stromal cell soluble/secreted factors should be followed-up.

R5) Following the reviewer’s suggestion, we have added a statement that various stromal cell soluble factors are needed to determine for more detailed study.

6) Line 632: Please explain the last sentence: “target molecules to inhibit?” I don’t understand this point. Perhaps explain more and expand.

R6) We rewrote the sentence to “Thus, trigger molecules for HIV-1 reactivation would be attractive drug targets for prevention of HIV spreading”.

Reviewer 2 Report

In this manuscript, Okutomi et al. evaluate molecular factors that contribute to HIV reactivation of latently infected cells. Stimulation of T cells and cell-to-cell contact is shown to be necessary to reactivate latent HIV-1. The virological synapse contributing to HIV-1 transfer through cell-to-cell contact is characterized through the illustration that the HIV-1 protein p17MA and gp120 colocalizes along with LFA-1 at contact sites. The authors elucidate that the LFA-1-ICAM-1 interaction as well as the TCR-CD3 complex both play a role in modulating latent HIV-1 reactivation. Several presumptive trigger molecules for reactivation of latency are also investigated, such as HLA class I. Finally, it is shown that live stromal cells are involved in modulating HIV-1 latency reactivation in the case of certain cell types. Overall the manuscript would be of interest to readers of viruses.  The experiments are well controlled and the authors provide significant evidence to support their conclusions.  I have the following corrections/comments to help improve the manuscript.

1. The authors put some emphasis in the text about the importance of their using autologous cells for their experiments compared to other studies that used allogeneic cells (dendritic cells, macrophages and so on).  If they want to make the point that this is a major issue for understanding the effects of co-culture on reactivation of latency then they should include experiments where they show that allogeneic T cell lines produce different effects compared to the autologous T cell lines used in this study. If they don’t provide data to support that the use of allogeneic cells could contribute to HIV reactivation then I am not sure that it is appropriate to make statements such as “implying that it possibly contributed to HIV-1 reactivation through cell stimulation” (line 539-40), or “However, considering that HIV-1 reactivation of latency occurs in infected individuals, studies should be performed in autologous cell coculture systems.” (line 62-64). 

2. Figure 4G,H could possibly use grey and dotted squares with a legend to describe the differences between the dotted and gray bars rather than only indicating the difference within the figure legend. The bottom panel stated as “virological synapse (%) in cell conjugates” is very unclear, these numbers do not correspond to any of the data bars in the graphs and the text as well as figure legend do not offer much better clarification.

3. Abstract: Line 23: “was significantly lower”, Line 25: “MHC Class I, that accumulated”

4. Introduction: Line 49 should read “… difficult to evaluate its in vivo contribution”,

5. Figure 3 is incorrectly titled as figure 1

6. Figure 4H typo at “virological synapse”

7. Figure 6C the labeling under the cytoplasm panel of Jurkat, Stim’d Jurkat, J1.1 does not align with the bands.

8. The acronym iTRAQ should be described in full as isoteric tags for relative and absolute quantitation during its first mention in the paper (the title of its appropriate section in the methods).

9. Discussion: Line 631 typo “from”, also “and stromal” should just be written as “stromal” in this sentence (delete the “and”).

Author Response

We appreciate the reviewer to consider our study interesting. The reviewer suggested the following points to clarify. We also thank for the reviewer’s kind English editing.

1) The authors put some emphasis in the text about the importance of their using autologous cells for their experiments compared to other studies that used allogeneic cells (dendritic cells, macrophages and so on). If they want to make the point that this is a major issue for understanding the effects of co-culture on reactivation of latency then they should include experiments where they show that allogeneic T cell lines produce different effects compared to the autologous T cell lines used in this study. If they don’t provide data to support that the use of allogeneic cells could contribute to HIV reactivation then I am not sure that it is appropriate to make statements such as “implying that it possibly contributed to HIV-1 reactivation through cell stimulation” (line 539-40), or “However, considering that HIV-1 reactivation of latency occurs in infected individuals, studies should be performed in autologous cell coculture systems.” (line 62-64).

R1) We have removed the descriptions pointed by the reviewer. Our suggestion that autologous cell coculture plays a role in reactivation of HIV latency is too early to be stated. We would like to emphasize that the stimulated status is important as signal donor cells rather than how the stimulated status is induced.

2) Figure 4G,H could possibly use grey and dotted squares with a legend to describe the differences between the dotted and gray bars rather than only indicating the difference within the figure legend. The bottom panel stated as “virological synapse (%) in cell conjugates” is very unclear, these numbers do not correspond to any of the data bars in the graphs and the text as well as figure legend do not offer much better clarification.

R2) We added the captions for gray and dotted bars to Figure 4G,H. The bottom panels in Figure 4G,H were changed to state as “virological synapse (%) in total cell conjugates”. The methods for calculating the efficiency of virological synapse formation (shown in the bottom panels) were added to in the figure legend. The numbers (shown in the bottom panels) mean that the virological synapse-positive cell conjugates relative to total cell conjugates.

3) Abstract: Line 23: “was significantly lower”, Line 25: “MHC Class I, that accumulated”.

R3) We appreciated the reviewer’s editing and corrected the text.

4) Introduction: Line 49 should read “… difficult to evaluate its in vivo contribution”,

R4) We corrected the text.

5) Figure 3 is incorrectly titled as figure 1.

R5) We apologize this error and have corrected the number of Figure suggested by the reviewer.

6) Figure 4H typo at “virological synapse”.

R6) The typo in Figure 4H was corrected.

7) Figure 6C the labeling under the cytoplasm panel of Jurkat, Stim’d Jurkat, J1.1 does not align with the bands.

R7) We corrected the position of the labeling for “cytoplasm” in Figure 6C.

8) The acronym iTRAQ should be described in full as isoteric tags for relative and absolute quantitation during its first mention in the paper (the title of its appropriate section in the methods).

R8) We have added “isobaric tag for relative and absolute quantitation” for the abbreviation iTRAQ to the method subtitle in the M&M (where it first appears).

9) Discussion: Line 631 typo “from”, also “and stromal” should just be written as “stromal” in this sentence (delete the “and”).

R9) We apologize our typos and corrected the sentence.

Reviewer 3 Report

This article addresses the role of cell contact in the activation of latent HIV. These studies have been performed in cell lines, which both limits the direct relevance but does allow for more mechanistic studies. The studies performed here are thorough and well-performed, and the authors clearly made their points. My comments are primarily (but not entirely) editorial in nature.

  1. Section 2.2: Cells are seeded at a rather high density for 5 day cultures. What is the cell count and viability as these cultures proceed?
  2. Throughout the manuscript the authors equate p24 concentration with HIV particle formation. They are not the same thing. P24 in HIV cultures is found in HIV particles, defective and partial particles, and even in its free form. Thus simply state that you are measuring p24 concentration, not the production of HIV particles.
  3. Figure 3 is erroneously labeled as figure 1.
  4. In lines 303-304 you summarize your data saying “TCR-CD3 expression on…signal donor cells did not impair cell conjugation but reduced the reactivation of latent HIV-1 in J1.1 ” I agree with this conclusion from the data. But this could undercut an important argument you are making, that contact results in HIV activation. But in this case it doesn’t, so a clearer argument should be made as to how this might be explained mechanistically.
  5. Figure 6B, the stars are hard to see. Can they be white or colored?
  6. The iTRAQ analyses (Table I) are interesting but essentially irrelevant. There’s no attempt to relate any of these proteins to the rest of the data. These analyses could be omitted from the manuscript.

Author Response

We appreciate the reviewer to consider our work well-performed. The reviewer suggested the followings to improve our study.

1) Section 2.2: Cells are seeded at a rather high density for 5 day cultures. What is the cell count and viability as these cultures proceed?

R1) We used relatively high cell density for cell-to-cell contact in our experiments. In J1.1 study, the cell coculture started at (1x106 Jurkat+1x106 J1.1)/2.1mL and the cell mixtures were observed at several time points post coculture. The approximate cell densities at 5 days post coculture were as follows: (1x106 Jurkat+2-3x106 J1.1)/2.1mL (when Jurkat were unstimulated) and (0.5x106 Jurkat+1x106 J1.1)/2.1mL (when Jurkat were stimulated). Thus, we found that i) when Jurkat were unstimulated, the cell density of the coculture increased 3-4-fold (most likely by J1.1 growth) but ii) when Jurkat were stimulated, the cell density of the coculture was slightly reduced (by the decrease/death of Jurkat because of PMA stimulation). Because the cell populations in the coculture were changed over time, we decided to evaluate HIV-1 reactivation based on the number of J1.1 cells at each time point.

2) Throughout the manuscript the authors equate p24 concentration with HIV particle formation. They are not the same thing. P24 in HIV cultures is found in HIV particles, defective and partial particles, and even in its free form. Thus simply state that you are measuring p24 concentration, not the production of HIV particles.

R2) We accepted the reviewer’s claim and modified throughout the text (Abstract, 2.3 section in M&M, 3.1 and 3.5 in Results, and the legends for Figure 1, 2, and 7).

3) Figure 3 is erroneously labeled as figure 1.

R3) We corrected the number of Figure suggested by the reviewer.

4) In lines 303-304 you summarize your data saying “TCR-CD3 expression on…signal donor cells did not impair cell conjugation but reduced the reactivation of latent HIV-1 in J1.1 ” I agree with this conclusion from the data. But this could undercut an important argument you are making, that contact results in HIV activation. But in this case it doesn’t, so a clearer argument should be made as to how this might be explained mechanistically.

R4) We greatly appreciate the reviewer’s suggestion. We have added the following text: “More importantly, the data suggest that cell contact alone was not sufficient for reactivation of latency in J1.1 cells and that subsequent signaling was required for the reactivation”.

5) Figure 6B, the stars are hard to see. Can they be white or colored?

R5) The stars were shown in light blue.

6) The iTRAQ analyses (Table I) are interesting but essentially irrelevant. There’s no attempt to relate any of these proteins to the rest of the data. These analyses could be omitted from the manuscript.

R6) We accepted the reviewer’s comment and removed Table I from the manuscript.